# PAY LESS ATTENTION WITH LIGHTWEIGHT AND DYNAMIC CONVOLUTIONS

**Felix Wu**[*]
Cornell University

**Angela Fan, Alexei Baevski, Yann N. Dauphin, Michael Auli**
Facebook AI Research

## ABSTRACT

Self-attention is a useful mechanism to build generative models for language and images. It determines the importance of context elements by comparing each element to the current time step. In this paper, we show that a very lightweight convolution can perform competitively to the best reported self-attention results. Next, we introduce dynamic convolutions which are simpler and more efficient than self-attention. We predict separate convolution kernels based solely on the current time-step in order to determine the importance of context elements. The number of operations required by this approach scales linearly in the input length, whereas self-attention is quadratic. Experiments on large-scale machine translation, language modeling and abstractive summarization show that dynamic convolutions improve over strong self-attention models. On the WMT'14 English-German test set dynamic convolutions achieve a new state of the art of 29.7 BLEU.[1]

## 1 INTRODUCTION

There has been much recent progress in sequence modeling through recurrent neural networks (RNN; Sutskever et al. 2014; Bahdanau et al. 2015; Wu et al. 2016), convolutional networks (CNN; Kalchbrenner et al. 2016; Gehring et al. 2016; 2017; Kaiser et al. 2017) and self-attention models (Paulus et al., 2017; Vaswani et al., 2017). RNNs integrate context information by updating a hidden state at every time-step, CNNs summarize a fixed size context through multiple layers, while as self-attention directly summarizes all context.

Attention assigns context elements *attention weights* which define a weighted sum over context representations (Bahdanau et al., 2015; Sukhbaatar et al., 2015; Chorowski et al., 2015; Luong et al., 2015). Source-target attention summarizes information from another sequence such as in machine translation while as self-attention operates over the current sequence. Self-attention has been formulated as *content-based* where attention weights are computed by comparing the current time-step to all elements in the context (Figure 1a). The ability to compute comparisons over such unrestricted context sizes are seen as a key characteristic of self-attention (Vaswani et al., 2017).

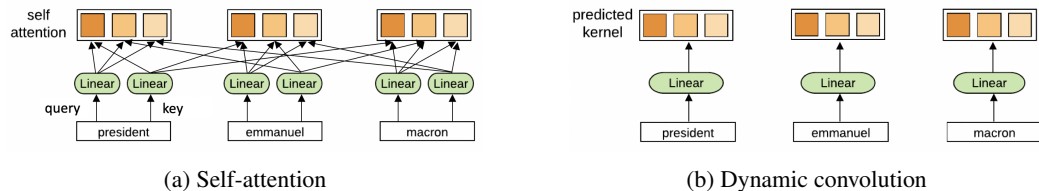

(a) Self-attention      (b) Dynamic convolution

Figure 1: Self-attention computes attention weights by comparing all pairs of elements to each other (a) while as dynamic convolutions predict separate kernels for each time-step (b).

However, the ability of self-attention to model long-range dependencies has recently come into question (Tang et al., 2018) and the unlimited context size is computationally very challenging due to the quadratic complexity in the input length. Furthermore, in practice long sequences require the introduction of hierarchies (Liu et al., 2018).

---

[*]Work done during an internship at Facebook.
[1]Code and pre-trained models available at `http://github.com/pytorch/fairseq`

In this paper, we introduce *lightweight convolutions* which are depth-wise separable (Sifre, 2014; Chollet, 2017; Kaiser et al., 2017), softmax-normalized and share weights over the channel dimension. The result is a convolution with several orders of magnitude fewer weights than a standard non-separable convolution. Different to self-attention, lightweight convolutions reuse the same weights for context elements, regardless of the current time-step.

*Dynamic convolutions* build on lightweight convolutions by predicting a different convolution kernel at every time-step. The kernel is a function of the current time-step only as opposed to the entire context as in self-attention (Figure 1b). Dynamic convolutions are similar to locally connected layers in the sense that the weights change at every position, however, the difference is that weights are dynamically generated by the model rather than fixed after training (LeCun et al., 1998; Taigman et al., 2014; Chen et al., 2015). Our approach also bears similarity to location-based attention which does not access the context to determine attention weights, however, we do not directly take the attention weights from the previous time-step into account (Chorowski et al., 2015; Luong et al., 2015). Shen et al. (2018b) reduce complexity by performing attention within blocks of the input sequence and Shen et al. (2017; 2018c) perform more fine-grained attention over each feature. Shen et al. (2018a) and Gong et al. (2018) use input-dependent filters for text classification tasks.

Our experiments show that lightweight convolutions perform competitively to strong self-attention results and that dynamic convolutions can perform even better. On WMT English-German translation dynamic convolutions achieve a new state of the art of 29.7 BLEU, on WMT English-French they match the best reported result in the literature, and on IWSLT German-English dynamic convolutions outperform self-attention by 0.8 BLEU. Dynamic convolutions achieve 20% faster runtime than a highly-optimized self-attention baseline. For language modeling on the Billion word benchmark dynamic convolutions perform as well as or better than self-attention and on CNN-DailyMail abstractive document summarization we outperform a strong self-attention model.

## 2 BACKGROUND

We first outline sequence to sequence learning and self-attention. Our work builds on non-separable convolutions as well as depthwise separable convolutions.

**Sequence to sequence learning** maps a source sequence to a target sequence via two separate networks such as in machine translation (Sutskever et al., 2014). The encoder network computes representations for the source sequence such as an English sentence and the decoder network autoregressively generates a target sequence based on the encoder output.

**The self-attention** module of Vaswani et al. (2017) applies three projections to the input $X \in \mathbb{R}^{n \times d}$ to obtain key (K), query (Q), and value (V) representations, where $n$ is the number of time steps, $d$ the input/output dimension (Figure 2a). It also defines a number of heads $H$ where each head can learn separate attention weights over $d_k$ features and attend to different positions. The module computes dot-products between key/query pairs, scales to stabilize training, and then softmax normalizes the result. Finally, it computes a weighted sum using the output of the value projection (V):

$$\text{Attention}(Q, K, V) = \text{softmax}(\frac{QK^T}{\sqrt{d_k}})V$$

**Depthwise convolutions** perform a convolution independently over every channel. The number of parameters can be reduced from $d^2k$ to $dk$ where $k$ is the kernel width. The output $O \in \mathbb{R}^{n \times d}$ of a depthwise convolution with weight $W \in \mathbb{R}^{d \times k}$ for element $i$ and output dimension $c$ is defined as:

$$O_{i,c} = \text{DepthwiseConv}(X, W_{c,:}, i, c) = \sum_{j=1}^{k} W_{c,j} \cdot X_{(i+j-\lceil \frac{k+1}{2} \rceil),c}$$

## 3 LIGHTWEIGHT CONVOLUTIONS

In this section, we introduce **LightConv**, a depthwise convolution which shares certain output channels and whose weights are normalized across the temporal dimension using a softmax. Compared to

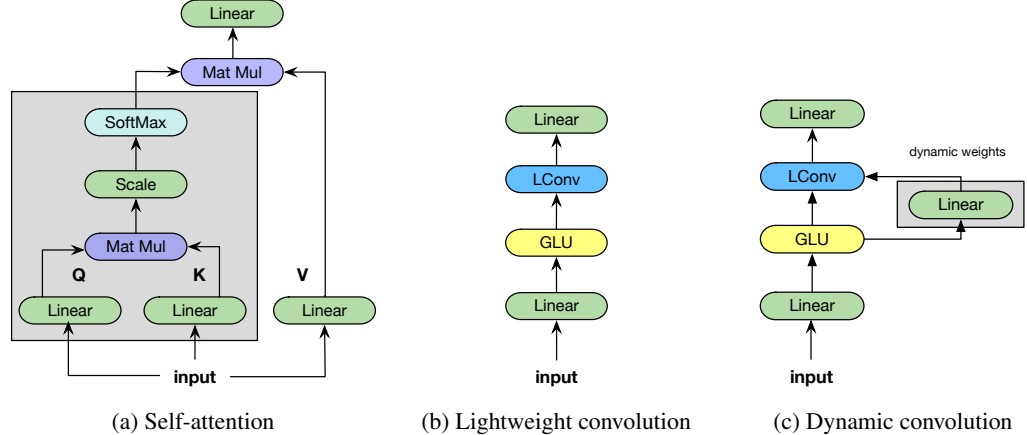

(a) Self-attention       (b) Lightweight convolution       (c) Dynamic convolution

Figure 2: Illustration of self-attention, lightweight convolutions and dynamic convolutions.

self-attention, LightConv has a fixed context window and it determines the importance of context elements with a set of weights that do not change over time steps. We will show that models equipped with lightweight convolutions show better generalization compared to regular convolutions and that they can be competitive to state-of-the-art self-attention models (§6). This is surprising because the common belief is that content-based self-attention mechanisms are necessary to obtaining state-of-the-art results in natural language processing applications. Furthermore, the low computational profile of LightConv enables us to formulate efficient dynamic convolutions (§4).

LightConv computes the following for the $i$-th element in the sequence and output channel $c$:

$$\text{LightConv}(X, W_{\lceil \frac{cH}{d} \rceil, :}, i, c) = \text{DepthwiseConv}(X, \text{softmax}(W_{\lceil \frac{cH}{d} \rceil, :}), i, c)$$

**Weight sharing.** We tie the parameters of every subsequent number of $\frac{d}{H}$ channels, which reduces the number of parameters by a factor of $\frac{d}{H}$. As illustration, a regular convolution requires 7,340,032 ($d^2 \times k$) weights for $d = 1024$ and $k = 7$, a depthwise separable convolution has 7,168 weights ($d \times k$), and with weight sharing, $H = 16$, we have only 112 ($H \times k$) weights. We will see that this vast reduction in the number of parameters is crucial to make dynamic convolutions possible on current hardware. Wang & Ji (2018) ties the weights of all channels (H = 1).

**Softmax-normalization.** We normalize the weights $W \in \mathbb{R}^{H \times k}$ across the temporal dimension $k$ using a softmax operation:

$$\text{softmax}(W)_{h,j} = \frac{\exp W_{h,j}}{\sum_{j'=1}^{k} \exp W_{h,j'}}$$

**Module.** Figure 2b shows the architecture of the module where we integrate LightConv. We first apply an input projection mapping from dimension $d$ to $2d$, followed by a gated linear unit (GLU; Dauphin et al. 2017), and the actual lightweight convolution. The GLU uses half of the inputs as gates by applying sigmoid units and then computes a pointwise product with the other inputs. We also apply an output projection of size $W^O \in \mathbb{R}^{d \times d}$ to the output of LightConv.

**Regularization.** We found DropConnect to be a good regularizer for the LightConv module (Wan et al., 2013). Specifically, we drop every entry of the normalized weights $softmax(W)$ with probability $p$ and divide it by $1 - p$ during training. This amounts to removing some of the temporal information within a channel.

**Implementation.** Existing CUDA primitives for convolutions did not perform very well to implement LightConv and we found the following solution faster on short sequences: We copy and expand the normalized weights $W \in \mathbb{R}^{H \times k}$ to a band matrix of size $BH \times n \times n$, where $B$ is the batch size. We then reshape and transpose the inputs to size $BH \times n \times \frac{d}{H}$, and perform a batch matrix multiplication to get the outputs. We expect a dedicated CUDA kernel to be much more efficient.

## 4 DYNAMIC CONVOLUTIONS

A dynamic convolution has kernels that vary over time as a learned function of the individual time steps. A dynamic version of standard convolutions would be impractical for current GPUs due to their large memory requirements. We address this problem by building on LightConv which drastically reduces the number of parameters (§3).

**DynamicConv** takes the same form as LightConv but uses a time-step dependent kernel that is computed using a function $f : \mathbb{R}^d \to \mathbb{R}^{H \times k}$:

$$\text{DynamicConv}(X, i, c) = \text{LightConv}(X, f(X_i)_{h,:}, i, c)$$

we model $f$ with a simple linear module with learned weights $W^Q \in \mathbb{R}^{H \times k \times d}$, i.e., $f(X_i) = \sum_{c=1}^{d} W_{h,j,c}^{Q} X_{i,c}$.

Similar to self-attention, DynamicConv changes the weights assigned to context elements over time. However, the weights of DynamicConv do not depend on the entire context, they are a function of the current time-step only. Self-attention requires a quadratic number of operations in the sentence length to compute attention weights, while the computation of dynamic kernels for DynamicConv scales linearly in the sequence length.

Our experiments (§6) show that models using DynamicConv match or exceed the performance of state-of-the-art models that use context-based self-attention. This challenges the typical intuitions about the importance of content-based self-attention in natural language processing applications.

## 5 EXPERIMENTAL SETUP

### 5.1 MODEL ARCHITECTURE

We use an encoder-decoder architecture for sequence to sequence learning (Sutskever et al., 2014) and we closely follow the architectural choices presented in Vaswani et al. (2017). Our self-attention baseline is the fairseq re-implementation of the Transformer Big architecture (Ott et al., 2018).[2]

The encoder and decoder networks have $N$ blocks each. Encoder blocks contain two sub-blocks: The first is a self-attention module (§2), a LightConv module (3), or a DynamicConv module (§4). The second sub-block is a feed-forward module: $ReLU(W^1 X + b_1)W^2 + b_2$ where $W^1 \in \mathbb{R}^{d \times d_{ff}}$, $W^2 \in \mathbb{R}^{d_{ff} \times d}$ and $d = 1024$, $d_{ff} = 4096$ unless otherwise stated. Sub-blocks are surrounded by residual connections (He et al., 2015) and layer normalization (Ba et al., 2016).

Decoder blocks are identical except that they have an additional source-target attention sub-block between the self-attention and feed-forward module. The source-target attention is equivalent to the self-attention module, except that the values and keys are projections over the encoder output for each source word.

Words are fed to the encoder and decoder networks in $d$ dimensional embeddings. We add sinusoidal position embeddings to encode the absolute position of each word in the sequence (Kaiser et al., 2017; Vaswani et al., 2017). The model computes a distribution over vocabulary $V$ by transforming the decoder output via a linear layer with weights $W^V \in \mathbb{R}^{d \times V}$ followed by softmax normalization.

LightConv and DynamicConv are identical to Transformer Big, except that self-attention modules are swapped with either fixed or dynamic convolutions. These models also use fewer parameters per block (cf. Figure 2b and Figure 2c) and we therefore increase the number of blocks to $N = 7$ for the encoder to roughly match the parameter count of Transformer Big. We generally set $H = 16$. Both LightConv and DynamicConv set the the encoder and decoder kernel sizes to 3, 7, 15, 31x4 for each block respectively; except for the decoder where we have only three top layers with kernel size 31.

### 5.2 DATASETS AND EVALUATION

To get a thorough understanding of the limitations of LightConv and DynamicConv we evaluate on three different tasks: machine translation, language modeling and abstractive summarization.

---

[2]https://github.com/pytorch/fairseq

**Machine Translation.** We report results on four benchmarks: For WMT English to German (En-De) we replicate the setup of Vaswani et al. (2017), based on WMT'16 training data with 4.5M sentence pairs, we validate on newstest2013 and test on newstest2014.[3] The vocabulary is a 32K joint source and target byte pair encoding (BPE; Sennrich et al. 2016). For WMT English to French (En-Fr), we borrow the setup of Gehring et al. (2017) with 36M training sentence pairs from WMT'14, validate on newstest2012+2013 and test on newstest2014. The 40K vocabulary is based on a joint source and target BPE factorization.

For WMT English to Chinese (Zh-En), we pre-process the WMT'17 training data following Hassan et al. (2018) resulting in 20M sentence pairs. We develop on devtest2017 and test on newstest2017. For IWSLT'14 German-English (De-En) we replicate the setup of Edunov et al. (2018) for 160K training sentence pairs and 10K joint BPE vocabulary. For this benchmark only, data is lowercased.

For WMT En-De, WMT En-Fr, we measure case-sensitive tokenized BLEU.[4] For WMT En-De only we apply compound splitting similar to Vaswani et al. (2017). For WMT Zh-En we measure detokenized BLEU to be comparable to Hassan et al. (2018).[5]

We train three random initializations of a each configuration and report test accuracy of the seed which resulted in the highest validation BLEU. Ablations are conducted on the validation set and we report the mean BLEU and standard deviation on this set. WMT En-De, WMT En-Fr are based on beam search with a beam width of 5, IWSLT uses beam 4, and WMT Zh-En beam 8 following Hassan et al. (2018). For all datasets, we tune a length penalty as well as the number of checkpoints to average on the validation set.

**Language Modeling.** We evaluate on the large-scale Billion word dataset (Chelba et al., 2013) which contains 768M tokens and has a vocabulary of nearly 800K types. Sentences in this dataset are shuffled and we batch sentences independently of each other. Models are evaluated in terms of perplexity on the valid and test portions.

**Summarization.** We test the model's ability to process long documents on the CNN-DailyMail summarization task (Hermann et al., 2015; Nallapati et al., 2016) comprising over 280K news articles paired with multi-sentence summaries. Articles are truncated to 400 tokens (See et al., 2017) and we use a BPE vocabulary of 30K types (Fan et al., 2017). We evaluate in terms of F1-Rouge, that is Rouge-1, Rouge-2 and Rouge-L (Lin, 2004).[6] When generating summaries, we follow standard practice in tuning the maximum output length, disallowing repeating the same trigram, and we apply a stepwise length penalty (Paulus et al., 2017; Fan et al., 2017; Wu et al., 2016).

## 5.3 Training and Hyperparameters

**Translation.** We use a dropout rate of 0.3 for WMT En-De and IWSLT De-En, 0.1 for WMT En-Fr, and 0.25 for WMT Zh-En. WMT models are optimized with Adam and a cosine learning rate schedule (Kingma & Ba, 2015; Loshchilov & Hutter, 2016) where the learning rate is first linearly warmed up for 10K steps from $10^{-7}$ to $10^{-3}$ and then annealed following a cosine rate with a single cycle. For IWSLT'14 De-En, we use a schedule based on the inverse square root of the current step (Vaswani et al., 2017). We train the WMT models on 8 NVIDIA V100 GPUs for a total of 30K steps on WMT En-De, 40K steps for WMT Zh-En and 80K steps for WMT En-Fr. For IWSLT De-En we train for 50K steps on a single GPU.

We use floating point 16 precision and accumulate the gradients for 16 batches before applying an update (Ott et al., 2018), except for IWSLT where we do not accumulate gradients. Batches contain up to 459K source tokens and the same number of target tokens for both WMT En-De and WMT Zh-En, 655K for En-Fr, and 4K for IWSLT De-En. We use label smoothing with $0.1$ weight for the uniform prior distribution over the vocabulary (Szegedy et al., 2015; Pereyra et al., 2017).

**Language Modeling.** We follow the same setup as for translation but remove the encoder module. For the Billion word benchmark we use an adaptive softmax output layer to reduce the computational burden of the large vocabulary (Grave et al., 2016; Press & Wolf, 2017) and tie it with variable sized

---

[3]`https://github.com/tensorflow/tensor2tensor/blob/321bacaa3abcca5dbf341ed6fb3d4a1531e513ff/tensor2tensor/data_generators/translate_ende.py#L60-L63`

[4]`https://github.com/moses-smt/mosesdecoder/blob/master/scripts/generic/multi-bleu.perl`

[5]SacreBLEU hash: `BLEU+case.mixed+lang.zh-en+numrefs.1+smooth.exp+test.wmt17+tok.13a+version.1.2.11`

[6]We use the following parameters for `ROUGE-1.5.5.pl`: -m -a -n 2

| Model | Param (En-De) | WMT En-De | WMT En-Fr |
|---|---|---|---|
| Gehring et al. (2017) | 216M | 25.2 | 40.5 |
| Vaswani et al. (2017) | 213M | 28.4 | 41.0 |
| Ahmed et al. (2017) | 213M | 28.9 | 41.4 |
| Chen et al. (2018) | 379M | 28.5 | 41.0 |
| Shaw et al. (2018) | - | 29.2 | 41.5 |
| Ott et al. (2018) | 210M | 29.3 | **43.2** |
| LightConv | 202M | 28.9 | 43.1 |
| DynamicConv | 213M | **29.7** | **43.2** |

Table 1: Machine translation accuracy in terms of BLEU for WMT En-De and WMT En-Fr on newstest2014.

| Model | Param (Zh-En) | IWSLT | WMT Zh-En |
|---|---|---|---|
| Deng et al. (2018) | - | 33.1 | - |
| Hassan et al. (2018) | - | - | 24.2 |
| Self-attention baseline | 292M | 34.4 | 23.8 |
| LightConv | 285M | 34.8 | 24.3 |
| DynamicConv | 296M | **35.2** | **24.4** |

Table 2: Machine translation accuracy in terms of BLEU on IWSLT and WMT Zh-En.

input word embeddings (Anonymous et al., 2018). The first 60K types in the adaptive softmax have dimension $1024$, the 100K types dimension $256$, and the last 633K types have size $64$.

We train on 32 GPUs with batches of 65K tokens for 975K updates. As optimizer we use Nesterov's accelerated gradient method (Sutskever et al., 2013) with a momentum value of $0.99$ and we re-normalize gradients if their norm exceeds $0.1$ (Pascanu et al., 2013). The learning rate is linearly warmed up from $10^{-7}$ to $1$ for 16K steps and then annealed using a cosine learning rate schedule (Loshchilov & Hutter, 2016) with one cycle.

**Summarization.** We train with Adam using the cosine learning rate schedule with a warmup of 10K steps and a period of 20K updates. We use weight decay 1e-3 and dropout 0.3.

## 6 RESULTS

### 6.1 MACHINE TRANSLATION

We first report results on WMT En-De and WMT En-Fr where we compare to the best results in the literature, most of which are based on self-attention. Table 1 shows that LightConv performs very competitively and only trails the state of the art result by 0.1 BLEU on WMT En-Fr; the state of the art is based on self-attention (Ott et al., 2018). This is despite the simplicity of LightConv which operates with a very small number of fixed weights over all time steps whereas self-attention computes dot-products with all context elements at every time-step.

DynamicConv outperforms the best known result on WMT En-De by 0.4 BLEU and achieves a new state of the art, whereas on WMT En-Fr it matches the state of the art. This shows that content-based self-attention is not necessary to achieve good accuracy on large translation benchmarks.

IWSLT is a much smaller benchmark and we therefore switch to a smaller architecture: $d_{ff} = 1024$, $d = 512$, and $H = 4$. The self-attention baseline on this dataset is the best reported result in the literature (Table 2).[7] LightConv outperforms this baseline by 0.4 BLEU and DynamicConv improves by 0.8 BLEU. We further run experiments on WMT Zh-En translation to evaluate on a non-European language. LightConv outperforms the baseline by 0.5 BLEU and DynamicConv by 0.6 BLEU.

---

[7] We omit comparison to Elbayad et al. (2018) since their test set is not directly comparable.

| Model | Param | BLEU | Sent/sec |
|---|---|---|---|
| Vaswani et al. (2017) | 213M | 26.4 | - |
| Self-attention baseline (k=inf, H=16) | 210M | $26.9 \pm 0.1$ | $52.1 \pm 0.1$ |
| Self-attention baseline (k=3,7,15,31x3, H=16) | 210M | $26.9 \pm 0.3$ | $54.9 \pm 0.2$ |
| CNN (k=3) | 208M | $25.9 \pm 0.2$ | $68.1 \pm 0.3$ |
| CNN Depthwise (k=3, H=1024) | 195M | $26.1 \pm 0.2$ | $67.1 \pm 1.0$ |
| + Increasing kernel (k=3,7,15,31x4, H=1024) | 195M | $26.4 \pm 0.2$ | $63.3 \pm 0.1$ |
| + DropConnect (H=1024) | 195M | $26.5 \pm 0.2$ | $63.3 \pm 0.1$ |
| + Weight sharing (H=16) | 195M | $26.5 \pm 0.1$ | $63.7 \pm 0.4$ |
| + Softmax-normalized weights [LightConv] (H=16) | 195M | $26.6 \pm 0.2$ | $63.6 \pm 0.1$ |
| + Dynamic weights [DynamicConv] (H=16) | 200M | $26.9 \pm 0.2$ | $62.6 \pm 0.4$ |
| Note: DynamicConv(H=16) w/o softmax-normalization | 200M | diverges | |
| AAN decoder + self-attn encoder | 260M | $26.8 \pm 0.1$ | $59.5 \pm 0.1$ |
| AAN decoder + AAN encoder | 310M | $22.5 \pm 0.1$ | $59.2 \pm 2.1$ |

Table 3: Ablation on WMT English-German newstest2013. (+) indicates that a result includes *all* preceding features. Speed results based on beam size 4, batch size 256 on an NVIDIA P100 GPU.

## 6.2 MODEL ABLATION

In this section we evaluate the impact of the various choices we made for LightConv (§3) and DynamicConv (§4). We first show that limiting the maximum context size of self-attention has no impact on validation accuracy (Table 3). Note that our baseline is stronger than the original result of Vaswani et al. (2017). Next, we replace self-attention blocks with non-separable convolutions (CNN) with kernel size 3 and input/output dimension $d = 1024$. The CNN block has no input and output projections compared to the baseline and we add one more encoder layer to assimilate the parameter count. This CNN with a narrow kernel trails self-attention by 1 BLEU.

We improve this result by switching to a depthwise separable convolution (CNN Depthwise) with input and output projections of size $d = 1024$. When we progressively increase the kernel width from lower to higher layers then this further improves accuracy. This narrows the gap to self-attention to only 0.5 BLEU. DropConnect gives a slight performance improvement and weight sharing does not decrease performance. Adding softmax normalization to the weights is only 0.3 BLEU below the accuracy of the baseline. This corresponds to LightConv. In Appendix A we compare softmax-normalization to various alternatives. Finally, dynamic convolutions (DynamicConv) achieve the same validation accuracy as self-attention with slightly fewer parameters and at 20% higher inference speed. Softmax-normalization is important for DynamicConv since training diverged in our experiments when removing it. To make the models more comparable, we do not introduce GLU after the input projection.

For comparison, we re-implemented averaged attention networks (AAN; Zhang et al. 2018) which compute a uniform average over past model states instead of a weighted average as in self-attention. Our re-implementation is efficient: we measure 129 sentences/sec for a base transformer-AAN on newstest2014 compared to 20 sentences/sec for Zhang et al. (2018). Table 3 shows that our models outperform this approach. Note that AANs still use self-attention in the encoder network while as our approach does away with self-attention both in the encoder and decoder.

## 6.3 LANGUAGE MODELING

As second task we consider language modeling on the Billion word benchmark. The self-attention baseline has $N = 16$ blocks, each with a self-attention module and a feed-forward module using $d_{ff} = 4096$ and $d = 1024$. DynamicConv uses $N = 17$ blocks to assimilate the parameter count and we use kernel sizes 15x2, 31x4 and 63x11. Table 4 shows that DynamicConv achieves slightly better perplexity than our self-attention baseline which is very competitive.

| Model | Param | Valid | Test |
|---|---|---|---|
| 2-layer LSTM-8192-1024 (Józefowicz et al., 2016) | – | – | 30.6 |
| Gated Convolutional Model (Dauphin et al., 2017) | 428M | – | 31.9 |
| Mixture of Experts (Shazeer et al., 2017) | 4371M [†] | – | 28.0 |
| Self-attention baseline | 331M | 26.67 | 26.73 |
| DynamicConv | 339M | 26.60 | **26.67** |

Table 4: Language modeling results on the Google Billion Word test set.
[†]does not include embedding and softmax layers

| Model | Param | Rouge-1 | Rouge-2 | Rouge-l |
|---|---|---|---|---|
| LSTM (Paulus et al., 2017) | - | 38.30 | 14.81 | 35.49 |
| CNN (Fan et al., 2017) | - | 39.06 | 15.38 | 35.77 |
| Self-attention baseline | 90M | 39.26 | 15.98 | 36.35 |
| LightConv | 86M | 39.52 | 15.97 | 36.51 |
| DynamicConv | 87M | **39.84** | **16.25** | **36.73** |
| RL (Celikyilmaz et al., 2018) | - | **41.69** | **19.47** | **37.92** |

Table 5: Results on CNN-DailyMail summarization. We compare to likelihood trained approaches except for Celikyilmaz et al. (2018).

## 6.4 ABSTRACTIVE SUMMARIZATION

Finally, we evaluate on the CNN-DailyMail abstractive document summarization benchmark where we encode a document of up to 400 words and generate multi-sentence summaries. This tests the ability of our model to deal with longer sequences. We reduce model capacity by setting $d = 1024$, $d_{ff} = 2048$, $H = 8$, similar to the Transformer base setup of Vaswani et al. (2017).

Table 5 shows that LightConv outperforms the self-attention baseline as well as comparable previous work and DynamicConv performs even better. We also show results for a reinforcement learning approach (Celikyilmaz et al., 2018) and note that RL is equally applicable to our architecture.[8]

## 7 CONCLUSION

We presented lightweight convolutions which perform competitively to the best reported results in the literature despite their simplicity. They have a very small parameter footprint and the kernel does not change over time-steps. This demonstrates that self-attention is not critical to achieve good accuracy on the language tasks we considered.

Dynamic convolutions build on lightweight convolutions by predicting a different kernel at every time-step, similar to the attention weights computed by self-attention. The dynamic weights are a function of the current time-step only rather than the entire context.

Our experiments show that lightweight convolutions can outperform a strong self-attention baseline on WMT'17 Chinese-English translation, IWSLT'14 German-English translation and CNN-DailyMail summarization. Dynamic convolutions improve further and achieve a new state of the art on the test set of WMT'14 English-German. Both lightweight convolution and dynamic convolution are 20% faster at runtime than self-attention. On Billion word language modeling we achieve comparable results to self-attention.

We are excited about the future of dynamic convolutions and plan to apply them to other tasks such as question answering and computer vision where inputs are even larger than the tasks we considered in this paper.

---

[8]An earlier version of this paper erroneously compared to Gehrmann et al. (2018), however, their setup is based on the full-text CNN-DailyMail whereas we use the more common entity-anonymized version.

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

SUPPLEMENTARY MATERIAL

## A    COMPARISON OF SOFTMAX-NORMALIZATION TO ALTERNATIVES

We compare our proposed softmax-normalization of weights to other alternatives in Table 6. For each setting, we use three seeds and report the mean and the standard deviation of the BLEU score on WMT English-German newstest2013. The softmax and norms are computed over the kernel dimension. Simply using the absolute value of the weights or squaring them does not make the training more stable, which shows that having all non-negative weights is not critical. Dividing the weights by the $\ell_2$-norm or bounding the weights with sigmoid or the hyperbolic tangent function also stablizes the training procedure; however, the softmax-normalization performs best.

| Method | BLEU |
|---|---|
| $W$ (No normalization) | diverges |
| $\text{softmax}(W)$ | $26.9 \pm 0.2$ |
| $\sigma(W)$ | $26.6 \pm 0.3$ |
| $\tanh(W)$ | $25.6 \pm 0.2$ |
| $\frac{W}{\|W\|_1+\epsilon}$ | diverges |
| $\frac{W}{\|W\|_2+\epsilon}$ | $26.8 \pm 0.2$ |
| $\text{power}(W, 2)$ | diverges |
| $\text{abs}(W)$ | diverges |
| $\frac{\text{abs}(W)}{\|W\|_1+\epsilon}$ | diverges |
| $\frac{\text{abs}(W)}{\|W\|_2+\epsilon}$ | $26.7 \pm 0.2$ |

Table 6: Alternatives to softmax-normalization in DynamicConv on WMT English-German newstest2013 ($\epsilon = 10^{-6}$).

## B    ON THE CURRENT STATE OF NON-AUTOREGRESSIVE GENERATION

In this section we compare DynamicConv to current non-autoregressive models in the literature. We measured generation speed for DynamicConv on a P100 GPU using batch size one to be comparable with other results. Results in the literature are based on either NVIDIA GTX-1080 GPUs or P100 GPUs. The effects of different GPU types is likely negligible because GPUs are vastly underutilized with batch size one.

Table 7 shows that DynamicConv with a single decoder layer outperforms all previously reported non-autoregressive results both in terms of speed as well as accuracy. Only two non-autoregressive concurrent efforts (Guo et al., 2019; Li et al., 2019) achieve a speedup over DynamicConv with a small drop in BLEU. Notably, both Guo et al. (2019) and Li et al. (2019) distill autoregressive models into non-autoregressive models (Hinton et al., 2015), in order to improve their results.

| Model (batch size = 1, beam size = 1) | Param | BLEU | Sent/sec | Tok/sec |
|---|---|---|---|---|
| NAT (+ FT) (Gu et al., 2018) | - | 17.7 | 25.6 | - |
| NAT (+ FT + NPD=10) (Gu et al., 2018) | - | 18.7 | 12.7 | - |
| NAT (+ FT + NPD=100) (Gu et al., 2018) | - | 19.2 | 3.9 | - |
| LT, Improved Semhash (Kaiser et al., 2018) | - | 19.8 | 9.5 | - |
| IR $i_{dec} = 1$ (Lee et al., 2018) | - | 13.9 | - | 511.4 |
| IR $i_{dec} = 2$ (Lee et al., 2018) | - | 17.0 | - | 393.6 |
| IR $i_{dec} = 5$ (Lee et al., 2018) | - | 20.3 | - | 139.7 |
| IR $i_{dec} = 10$ (Lee et al., 2018) | - | 21.6 | - | 90.4 |
| IR Adaptive (Lee et al., 2018) | - | 21.5 | - | 107.2 |
| NART w/ hints (Li et al., 2019) | - | 21.1 | 38.5 | - |
| NART w/ hints ($B = 4$, 9 candidates) (Li et al., 2019) | - | 25.2 | 22.7 | - |
| ENAT Embedding Mapping (Guo et al., 2019) | - | 20.7 | 41.7 | - |
| ENAT Embedding Mapping (rescoring 9 candidates) (Guo et al., 2019) | - | 24.3 | 20.4 | - |
| Autoregressive (Gu et al., 2018) | - | 22.7 | 2.5 | - |
| Autoregressive (Lee et al., 2018) | - | 23.8 | - | 54.0 |
| Transformer (Li et al., 2019) | - | 27.3 | 1.3 | - |
| Transformer (Guo et al., 2019) | - | 27.4 | 1.6 | - |
| DynamicConv (1-decoder layer (k=31)) | 124M | 26.1 | 15.2 | 423.0 |
| DynamicConv (3-decoder layers (k=3,7,15)) | 153M | 27.7 | 7.2 | 202.3 |
| DynamicConv (6-decoder layers (k=3,7,15,31,31,31)) | 200M | 28.5 | 3.9 | 110.9 |

Table 7: Inference speed of non-autoregressive models and small decoder versions of DynamicConv on WMT English-German newstest2014. For some models, the decoding speed (sent/sec) is derived by taking the inverse of the sentence generation latency in the literature.

