# OpenReview forum: "Pay Less Attention with Lightweight and Dynamic Convolutions"
_ICLR.cc/2019/Conference_

### Official Review · AnonReviewer3 · 2018-11-02
**well-written, surprising and promising results**

**Rating:** 8
**Confidence:** 4

**Review:**

The paper proposes a convolutional alternative to self-attention. To achieve this, the number of parameters of a typical convolution operation is first reduced by using a depth-wise approach (i.e. convolving only within each channel), and then further reduced by tying parameters across layers in a round-robin fashion. A softmax is applied to the filter weights, so that the operation computes weighted sums of its (local) input (LightConv).

Because the number of parameters is dramatically reduced now, they can be replaced by the output of an input-dependent linear layer (DynamicConv), which gives the resulting operation a "local attention" flavour. The weights depend only on the current position, as opposed to the attention weights in self-attention which depend on all positions. This implies that the operation is linear in the number of positions as opposed to quadratic, which is a significant advantage in terms of scaling and computation time.

In the paper, several NLP benchmarks (machine translation, language modeling) that were previously used to demonstrate the efficacy of self-attention models are tackled with models using LightConv and DynamicConv instead, and they are shown to be competitive across the board (with the number of model parameters kept approximately the same).

This paper is well-written and easy to follow. The proposed approach is explained and motivated well. The experiments are thorough and the results are convincing. I especially appreciated the ablation experiment for which results are shown in Table 3, which provides some useful insights beyond the main point of the paper. The fact that a linear time approach can match the performance of self-attention based models is a very promising and somewhat surprising result.

In section 5.3, I did not understand what "head band, next band, last band" refers to. I assume this is described in the anonymous paper that is cited, so I suppose this is an artifact of blind review. Still, even with the reference unmasked it might be useful to add some context here.

---

> ### Comment · AnonReviewer1 · 2018-11-06
> **Where "head band" comes from**
>
> The "head band, next band, last band" terminology is from https://openreview.net/forum?id=ByxZX20qFQ, which is presumably the cited anonymous paper.

---

> ### Author Response · Authors · 2018-11-17
> **Response to Reviewer #3**
>
> Thank you for your comments. We improved the description of the adaptive softmax hyperparameters ('band' terminology) in the updated version of the paper. We hope this is clearer now.
>
> We refer to different subsets of the vocabulary as 'bands'. The most frequent words are denoted as "head band", and so on.

---

### Official Review · AnonReviewer2 · 2018-11-05
**Interesting work, strong results, good paper**

**Rating:** 8
**Confidence:** 4

**Review:**

Overall, this is a really good paper.
The authors propose an alternative to content based similarity for NL applications as compared to self-attention models by proposing the parameter and sequence length efficient Lightweight and Dynamic Convolutions.
The authors show, over various NL tasks like Translation, LM and Abstractive summarisation, the comparison of self attention models with Lightweight and Dynamic convolution layer.
The weight sharing was particularly interesting and can be seen as applying different heads for the same kernel.

The experimental results give strong evidence for these alternatives proposed by the authors.
The lightweight and dynamic convolution layers, both perform similar or better than the self-attention layer in all the tasks.
The WMT EnFr result is much better than all the other models, establishing a new state of the art.

Question for the authors:
1. Is the weight sharing within the kernel mostly for reducing computation?
If so, did you trying varying H size and measure how much that affects performance? What is surprising is that, in the ablation table the weight sharing increases the BLEU score by 0.1.
2. Did you run any experiments where the kernel size covers the whole sentence?
3. Since the number of parameters only change linearly wrt sequence length, did you try running this on datasets that have really long sequences to show the effectiveness of this approach further?
4. How important was softmax normalization for training?

---

> ### Author Response · Authors · 2018-11-17
> **Response to Reviewer #2**
>
> Thank you for your fruitful comments.
> Q1: For DynamicConv, weight sharing reduces both computation and memory footprint, while for LightConv, it only reduces memory footprint.  Yes, we did try using large H sizes; however, the performance degrades and the memory footprint increases dramatically which prohibits us from using a large batch size. As a consequence, training becomes much slower.  For your information, DynamicConv with H=64 gets BLEU score 26.8 ± 0.1 on newstest2013 compared to 26.9 ± 0.2 with H=16 in Table 3.
>
> Q2: We conducted an additional experiment based on your suggestion. We set the encoder kernel size to 237 and the decoder kernel size to 267 at each layer to cover the whole sequence. The BLEU score drops slightly to 26.7 ± 0.1. This is a small difference and we expect that slightly tuned hyperparameters would close the gap.
>
> Q3: In section 6.4, we show experiments for document summarization (CNN/DailyMail) where the input sequence is capped at 400 words and the output sequence is 57 words on average with some examples having summaries of up to 478 words. Our results show that the model performs very well in this setting.
>
> Q4: We found it very important as training diverged without softmax-normalization (see Note in Table 3) for DynamicConv. We added a comparison of softmax-normalization to various alternatives to Appendix A of the updated paper.
> Furthermore, we are able to train the model without softmax-normalization with more aggressive gradient clipping, a lower learning rate (reducing it by a factor of 5) and more updates (increasing it by 5 times), but this slowed down training dramatically.

---

### Official Review · AnonReviewer1 · 2018-11-06
**Major advance in sequence-to-sequence architectures**

**Rating:** 8
**Confidence:** 4

**Review:**

The authors present lightweight convolutions and dynamic convolutions, two significant advances over existing depthwise convolution sequence models, and demonstrate very strong results on machine translation, language modeling, and summarization. Their results go even further than those of the Transformer paper in countering the conventional wisdom that recurrence (or another way of directly modeling long-distance dependencies) is crucial for sequence-to-sequence tasks. Some things that I noticed:

- While you do cite "Depthwise Separable Convolutions for Neural Machine Translation" from Kaiser et al. (ICLR 2018), there are some missed opportunities to compare more directly to that paper (e.g., by comparing to their super-separable convolutions). Kaiser et al. somewhat slipped under the community's radar after the same group released the Transformer on arXiv a week later, but it is in some ways a more direct inspiration for your work than the Transformer paper itself.

- I'd like to see more analysis of the local self-attention ablation. It's fantastic to see such a well-executed ablation study, especially one that includes this important comparison, but I'd like to understand more about the advantages and drawbacks of local self-attention compared to dynamic convolutions. (For instance, dynamic convolutions are somewhat faster at inference time in your results, but I'm unsure if this is contingent on implementation choices or if it's inherent to the architecture.)

- From a systems and implementation perspective, it would be great to see some algorithm-level comparisons of parallelism and critical path length between dynamic convolutions and self-attention. My gut feeling is that dynamic convolutions significantly more amenable to parallelization on certain kinds of hardware, especially at train time, but that the caching that's possible in self-attention inference might make the approaches more comparable in terms of critical path latency at inference time; this doesn't necessarily line up with your results so far though.

- You mostly focus on inference time, but you're not always as clear about that as you could be; I'd also like to see train time numbers. Fairseq is incredibly fast on both sides (perhaps instead of just saying "highly optimized" you can point to a paper or blog post?)

- The nomenclature in this space makes me sad (not your fault). Other papers (particularly a series of three papers from Tao Shen at University of Technology Sydney and Tianyi Zhou at UW) have proposed architectures that are similarly intermediate between self-attention and (in their case 1x1) convolution, but have decided to call them variants of self-attention. I could easily imagine a world where one of these groups proposed exactly your approach but called it "Dynamic Local Self-Attention," or even a world where they've already done so but we can't find it among the zillions of self-attention variants proposed in the past year. Not sure if there's anything anyone can do about that, but perhaps it would be helpful to briefly cite/compare to some of the Shen/Zhou work.

- I think you should have tried a language modeling dataset with longer-term dependencies, like WikiText-103. Especially if the results were slightly weaker than Transformer, that would help place dynamic convolutions in the architecture trade-off space.

That last one is probably my most significant concern, and one that should be fairly easy to address. But it's already a great paper.

---

> ### Author Response · Authors · 2018-11-17
> **Response to Reviewer #1**
>
> Thank you for your fruitful comments.
> Q: Comparison to "Depthwise Separable Convolutions for Neural Machine Translation" from Kaiser et al. (ICLR 2018).
> Super-separable convolutions modify the pointwise operation (introducing groups) that follows depthwise convolutions, while we modify the latter (which focuses on aggregating the temporal information). We did try to introduce groups to the linear layers of the feed-forward block (that follows light/dynamic convolutions) but to reach the same accuracy, we had to increase the number of parameters of the model to a similar level as with the dense linear, at which point the network became slower.
>
> Q: “I'd also like to see train time numbers”
> Here are training times:
> Self-attention, 17.5h (with or without limited window)
> DynamicConv, 16.9h
> LightConv, 16.5h
>
> Our current implementation of dynamic convolutions is actually quite inefficient. We put the various convolution kernels in a sparse tensor that has only non-zero entries for the diagonal entry, thus using a lot of space. We expect a dedicated CUDA kernel to be more efficient. We are investigating such a kernel.
>
> Note that batching is much more efficient during training which smooths out some of the speed advantages we see at test time. During inference batching is by far not as efficient due to repeated invocation of the decoder at every time step.
>
> Q: “Other papers (particularly a series of three papers from Tao Shen at University of Technology Sydney and Tianyi Zhou at UW) have proposed architectures that are similarly intermediate between self-attention and (in their case 1x1) convolution”
> We will discuss our work in the light of Shen & Zhou (2017, 2018) and also reference Ott et al. (2018) wrt fairseq speed.
>
> Q: “You should have tried a language modeling dataset with longer-term dependencies”
> In section 6.4, we show experiments for CNN/DailyMail document summarization which entails long input and output sequences. The input sequence is capped at 400 words and the output sequence is 57 words on average with some examples having summaries of up to 478 words.

---

### Public Comment · (anonymous) · 2018-11-08
**Hi, the Code link is not available!**

Hi, the Code link is not available!

---

> ### Author Response · Authors · 2018-11-17
> **code**
>
> We are planning to share the code later.

---

### Public Comment · (anonymous) · 2018-11-13
**Hi can you explain the advantages over bytenet**

Hi,

I have a question. You claim that your lightweight cnn can has fewer parameters and linear time. I think it is very necessary to compare with a well-know CNN sequence baseline, i.e. bytenet. it is also a pure con sequence model and shows very good performance in language modeling and translation. Have you compare with it?? Better accuracy or higher efficiency??

Do you plan to you share your code? I am quite interested.

---

> ### Author Response · Authors · 2018-11-17
> **Re: explain the advantages over bytenet**
>
> We do compare to a CNN baseline (non-separable convolutions), see Table 3 "CNN (k=3)". However, our model does use source-target attention which is not the case for ByteNet. Finally, our model performs better on newstest2014 of WMT English-German translation at 29.7 BLEU vs. 23.75 BLEU for ByteNet.
>
> And yes, we will release the code.

---

> > ### Public Comment · (anonymous) · 2018-11-30
> > **hi**
> >
> > please note bytenet can also be used for language model, i.e., using  only decoder. So it is very important to compare with it, which is also one type of lightweight cnn

---

### Public Comment · (anonymous) · 2018-11-13
**can you explain this**

1 what do you mean by saying “We expect a dedicated CUDA kernel to be much more efficient.”

You mean the efficiency. advantage in current CUDA is not obvious??

is it possible to expect a new CUDA kernel specifically designed for your model

2 code is not available

Code and pre-trained models available at http://anonymized

---

> ### Author Response · Authors · 2018-11-17
> **CUDA kernel & code**
>
> We are currently investigating a dedicated CUDA kernel and we will make the code available.

---

### Public Comment · (anonymous) · 2018-11-15
**Dear authors,**

I found this paper is very interesting, would you like to share the source code, which is very helpful for fully understanding it

---

> ### Author Response · Authors · 2018-11-17
> **Open sourcing the code**
>
> Yes, we will share the code at a later stage!

---

### Public Comment · (anonymous) · 2018-12-07
**Can you explain the difference between your work with more closely related works such as Convolutional Net (Gong et al.) and Context-Sensitive Convolution (Shen et al.)**

I found your work very interesting, but there are some recent works that are closely related to your work, which take a sentence as input and generate convolutional kernels that are further applied on the sentence, but with a different granularity. I think those works are definitely worth comparing to.

missing references:
Learning Context-Sensitive Convolutional Filters for Text Processing (Shen et al.)
Convolutional Interaction Network for Natural Language Inference (Gong et al.)

---

> ### Author Response · Authors · 2018-12-27
> **Thank you for pointing out these interesting CNN papers!**
>
> There are indeed some similarities to their work, but there are also significant differences, including:
> 1) Their methods focus on using the information from one sequence to generate the convolution filter that operates on the other sequence, while we focus on using one sequence as both the source and the target like self-attentions. Admittedly, Gong et al. use intra-sentence convolutional interactions; however, their ablation study is limited to removing them instead of replacing them with self-attentions.
> 2) They use a filter generator network to predict the kernel, while DynamicConv requires only a simple linear projection.
> 3) Their models use the same convolution filter at each time step, while our DynamicConv uses different filters at each time step. This is possible due to LightConv (depthwise + weight sharing) which significantly reduces the number of parameters.
> 4) Their filter generator network uses the information from the whole sequence to generate a convolutional filter, while we only use the information at the current time step.
> 5) Our filters are softmax-normalized, while theirs are not.
>
> We consider their work as orthogonal to our methods. Future work may try to apply LightConv and DynamicConv to their models in order to achieve even better performance!

---

### Public Comment · (anonymous) · 2018-12-31
**Problems on the implementation of LightConv**

It's a good paper and easy to catch, and I got 2 problems confused me when reading it. Forgive me if I misunderstand the paper.

1. In the weight sharing of LightConv in Section 3, as far as I understand, the description "We tie the parameters of every subsequent number of d/H channels, which reduces the number of parameters by a factor of H." should led to 448 (d/H x  k) weights, instead of 112 stated, which I think is a typo.

2. As for the implementation in the same section, the operation "batch matrix multiplication" confused me a lot. As LightConv is a conv operator, we can implement it by (1) applying image_to_coloum to input, (2) copying the kernels and (3) taking matrix multiplication. These procedures are taken by many DL platforms, like Caffe, to implement the conv operator. But the paper states that only reshape and transpose are applied to input before "batch matrix multiplication", which seems an aggregation over all position (same sprit as self-attention) when taking batch matrix multiplication, conflicting to the paper's claim.


Looking forward to your code!

---

> ### Author Response · Authors · 2019-01-07
> **Re: Problems on the implementation of LightConv**
>
> 1. Thank you for pointing out this typo.  It should reduce the number of parameters by a factor of “d/H” rather than “H”, so 112 is still the correct number.
>
> 2. The matrix would be a band matrix, i.e. entries outside of the kernel are zeros. As you noticed, this is similar to the matrix multiplication in the self-attention, which has O(n^2) time complexity. However, we observe that when the sequence is short (< 1000), this implementation is practically faster.

---

### Public Comment · (anonymous) · 2019-01-15
**Problem about lightconv**

I'm not sure if I have fully-understand your great work.
In my opinion, the difference between you and (Kaiser et al., 2017) is the softmax-normalized and share weights over the channel dimension. And you use these two mechanism not only reduce the number of parameter, but increase the result greatly(from 26.1 to 28.9).
So can you explain why these two mechanism is so useful?
Thanks

---

> ### Author Response · Authors · 2019-01-16
> **Other components besides the convolutions and self-attentions are also important**
>
> One of our contributions is to better understand the importance of self-attention which is often perceived as the most important design choice in the architecture of Vaswani et al.
> Table 3 of our paper shows that self-attention alone only accounts for a small portion of the improvement of Vaswani et al. over previous work, e.g., row 6 in Table 3 "CNN Depthwise + Increasing kernel" uses the depthwise convolution of Kaiser et al. (2017) and it is only 0.5 BLEU behind our reimplementation of Vaswani et al.
> Therefore, modeling choices other than self-attention contribute a very large fraction of the improvement of Vaswani et al. over other work. In early experiments, we found that FFN blocks between the self-attention module in the Transformer are very important.

---

### Meta-Review · Area_Chair1 · 2018-12-02
**Accept**

**Confidence:** 4
**Recommendation:** Accept (Oral)

**Metareview:**

Very solid work, recognized by all reviewers as worthy of acceptance. Additional readers also commented and there is interest in the open source implementation that the authors promise to provide.